# Combined Permutation Tests for Pairwise Comparison of Scale Parameters Using Deviances

**Scott J. Richter** [1,*] **and Melinda H. McCann** [2]

[1] Department of Mathematics and Statistics, University of North Carolina at Greensboro, Greensboro, NC 27402, USA
[2] Department of Statistics, Oklahoma State University, Stillwater, OK 74075, USA; mindy.mccann@okstate.edu
[*] Correspondence: sjricht2@uncg.edu

**Abstract:** Nonparametric combinations of permutation tests for pairwise comparison of scale parameters, based on deviances, are examined. Permutation tests for comparing two or more groups based on the ratio of deviances have been investigated, and a procedure based on Higgins' RMD statistic was found to perform well, but two other tests were sometimes more powerful. Thus, combinations of these tests are investigated. A simulation study shows a combined test can be more powerful than any single test.

**Keywords:** deviance; scale parameter; pairwise comparison; permutation test; combined test

## 1. Introduction

Tests for homogeneity of scale are of interest in many areas of application, including industrial quality assurance, agricultural production and education [1]. Parametric tests for comparing scale (e.g., [2–4]) are generally not robust to nonnormality (see [5]). Consequently, more robust alternatives are of interest.

An approximate test using the ANOVA *F*-test on the absolute deviations from the mean was proposed [6]. Using absolute deviations from the median (referred to as deviances in the remainder of this paper), referred to as the *W50* test, was later suggested [7]. However, no uniformly best test for scale has been demonstrated in the literature. In fact, without more stringent distributional assumptions, the minimal sufficient statistic would generally be the n-dimensional vector of order statistics. Thus, no single statistic exists that summarizes the information contained in the data, and a uniformly best test statistic does not generally exist. In spite of this, the *W50* test has been recommended as a computationally simple test showing good overall performance with respect to power and robustness to nonnormality in several comparative studies ([8–10]). More recently, a study [5] compared 25 omnibus tests for homogeneity of variance and recommended the *W50* test as "superior". A modification of Levene's test [6] (referred to as *OB*) was proposed [11] which has been recommended over the *W50* test for light-tailed distributions [12]. The *W50* and *OB* tests, as well as permutation versions of these tests, were evaluated [5] and it was found that the permutation versions tended to be more robust and have higher power. The *W50* test was recommended as a computationally simple robust test, as was the permutation version of the *OB* test for symmetric and lighter-tailed skewed distributions. Another test for scale utilizing deviances, based on the ratio of the mean deviances, was also proposed [13]. This test will be referred to as the *RMD* test. The *RMD* test was found [14] to be generally superior to *W50* and *OB*, although there were still cases where each of *W50* and *OB* had higher power. Since no test has been found to be uniformly superior, it is of interest to develop a test that combines these three tests. A combined test of scale parameters based on the IQR was studied [15] and the combined test was found to be more powerful than its constituent tests in some scenarios. Similarly, we will investigate

nonparametric combinations of the *RMD*, *W50* and *OB* tests to determine if combining the tests can provide increased power compared to individual tests.

## 2. Methods for Comparing Scale Parameters

Consider a one-way layout with $t$ treatments and $n_i$ observations per treatment. We assume a location-scale model, $y_{ij} = \mu_i + \sigma_i \varepsilon_{ij}$, $i = 1, \ldots, t$, $j = 1, \ldots, n_i$, where $\mu_i$ and $\sigma_i$ are the location and scale parameters, respectively, of treatment $i$, and $\varepsilon_{ij}$ are independent and identically distributed with median 0. It is desired to test $H_0 : \sigma_1 = \sigma_2 = \cdots = \sigma_t$ versus $H_a : \sigma_i \neq \sigma_j$ for some $i$ and $j$.

### 2.1. Brown–Forsythe (W50) Test

First, compute the deviances, $\widetilde{z}_{ij} = \left| y_{ij} - \widetilde{y}_i \right|$, where $\widetilde{y}$ is the sample median. The ANOVA *F* test is performed on these scores, and the *p*-value is based on the *F* distribution with $t - 1$ and $n - t$ degrees of freedom [7].

### 2.2. Higgins' (RMD) Test

The statistic is defined as, $RMD = \dfrac{\max\left( \overline{\widetilde{z}}_i, \overline{\widetilde{z}}_j \right)}{\min\left( \overline{\widetilde{z}}_i, \overline{\widetilde{z}}_j \right)}$, where $\overline{\widetilde{z}}_i$ is the mean of the deviances, $\widetilde{z}_{ij}$, for treatment $i$. The deviances $\widetilde{z}_{ij} = \left| y_{ij} - \widetilde{y}_i \right|$ are the same as those used by the *W50* test. The permutation distribution of the *RMD* statistic was used to calculate a *p*-value [13].

### 2.3. O'Brien's (OB) Test

First, compute the scores $r_{ij}(w) = \dfrac{\left[ (w + n_i - 2) n_{ij} \left( y_{ij} - \overline{y}_i \right)^2 - w s_i^2 (n_i - 1) \right]}{[(n_i - 1)(n_i - 2)]}$, where $0 \leq w \leq 1$. At one extreme, when $w = 0$, the statistic reduces to $r_{ij}(0) = \dfrac{n_i \left( y_{ij} - \overline{y}_i \right)^2}{n_i - 1}$, which is a slight modification of Levene's test, which uses $\overline{z}_{ij}^2 = \left( y_{ij} - \overline{y}_i \right)^2$. At the other extreme, when $w = 1$, $r_{ij}(1) = q_{ij} = \dfrac{\left[ n_i \left( y_{ij} - \overline{y}_i \right)^2 - s_i^2 \right]}{n_i - 2} = n_i s_i^2 - (n_i - 1) s_{i-1}^2$, which was referred to as a "jackknife pseudovalue of $s_i^2$ [11]". The ANOVA F test is performed on these scores and, the *p*-value is based on the *F* distribution with $t - 1$ and $n - t$ degrees of freedom. Tests based on $\overline{z}_{ij}^2$ have been shown to have inflated Type I error rates, while those based on $q_{ij}$ tend to have low power. Since $r(w)$ is a weighted average of the two tests, it provides a way to balance the drawbacks of the two tests. A "utility" value of $w = 0.5$ was suggested for most situations [11], and this is the value employed in this study.

### 2.4. Permutation Tests

While the permutation test using the *RMD* statistic was suggested [13], the *W50* and *OB* tests described previously were proposed as approximate tests based on the *F* distribution. However, *p*-values for *W50* and *OB* can also be calculated using permutation distributions. A simulation study [1] found for the two-treatment case that the permutation versions tended to be more robust and have greater power than the approximate tests. Thus, we will consider only the permutation versions of these combined tests. Test statistics will be computed for a large number of random reassignments of observations to treatments, and the *p*-value will be calculated as the proportion of values of the permutation distribution that is at least as extreme as the observed test statistic value.

## 3. Combined Tests

A two-step approach to create a nonparametric combination of dependent tests was proposed [16] and described as follows:

Step 1. Analyze the data using the tests of interest, referred to as partial tests;

Step 2. Combine the partial tests to assess the global hypothesis.

Several different combining functions have been developed that satisfy the properties required for a suitable combining function [16]. Since the relative power of different combining functions can vary across conditions, we consider combined tests using three of the best-known combining functions: the Fisher, Liptak and Tippett combining functions [15].

Let $\lambda_i$ be the $p$-value associated with the $i$th test to be combined. Then, the test statistics for the Fisher, Liptak and Tippett functions are

1.  The Fisher combining function is $T_F = -\sum_i \ln(\lambda_i)$;
2.  The Liptak combining function is $T_L = \sum_i \Phi^{-1}(1 - \lambda_i)$;
3.  The Tippett combining function is $T_T = \max_i(1 - \lambda_i)$.

The Tippett function tends to have the highest power when one or a few, but not all, of the constituent tests reject the null hypothesis; the Liptak function tends to have the highest power when all tests reject the null hypothesis; the power of the Fisher function will tend to lie between the other two, making it the more general option and thus probably the most popular [16]. The combined tests are carried out as follows [16].

1.  Compute the observed test statistic value ($T_F$, $T_L$, $T_T$) according to the above definitions, using the permutation $p$-values of *RMD*, *W50* and *OB*.
2.  To compute the permutation test $p$-value associated with each combined statistic:

    i   For the $i$th statistic in the permutation distributions constructed for *RMD*, *W50* and *OB*, compute the $i$th partial $p$-value as the proportion of test statistic values at least as large as the $i$th statistic value.

    ii  Using the partial $p$-values for *RMD*, *W50* and *OB*, use the respective combining function to compute a test statistic value ($T_F$, $T_L$, $T_T$) for each permutation. This results in a permutation distribution for each of the combined statistics.

    iii For each combined test, the permutation $p$-value is then the proportion of values in the permutation distribution at least as large as the observed statistic value.

Note that all tests are based on the same set of randomly generated permutations.

Since the *RMD*, *W50* and *OB* tests were each most powerful for at least some scenarios in past simulations (e.g., [5]), combinations of these three tests will be examined. In addition, since *RMD* and *W50* were usually more powerful than *OB*, a combination of only *RMD* and *W50* will also be considered. The $p$-values for each of the constituent tests in each combination will be estimated using the permutation distribution of the statistic. The powers Type I error rates of the Fisher, Liptak and Tippett combining functions will be estimated and compared, and these will also be compared to those of the individual tests.

## 4. Strong Familywise Error Rate Control for Pairwise Comparisons

The familywise error rate (FWER) will be controlled using the technique of Richter and McCann [17]. Richter and McCann [17] proposed a restricted permutation method to provide strong control of the familywise error rate (FWER) for pairwise comparison of location parameters. This method will be extended to the present case of comparing scale parameters as follows. First, the two-sample test statistic for a given method will be calculated for each of the possible $t(t-1)/2$ pairs of treatments. Then, the maximum value of the test statistic across all pairs will be calculated. Next, observations will be reassigned at random to treatments within each pair of treatments, a test statistic calculated for each pair of treatments, and the maximum value determined. This will be repeated many times to build the permutation distribution, and the $p$-value for comparing each pair of treatments will be calculated as the proportion of values in the permutation distribution that is at least as extreme as the observed value.

## 5. Simulation Study

*5.1. Procedures Studied*

A simulation study estimated and compared the familywise Type I error rate and "any-pair" power (probability of detecting at least one true difference) of the methods described in Section 2:

1. *RMD*: Higgins *RMD* procedure.
2. *W50*: Brown and Forsyth's *W50* test.
3. *OB*: O'Brien's method using means.
4. $F_3$ : Fisher's combination test of *RMD*, *W50* and *OB*.
5. $F_2$ : Fisher's combination test of *RMD* and *W50*.
6. $L_3$ : Liptak's combination test of *RMD*, *W50* and *OB*.
7. $L_2$ : Liptak's combination test of *RMD* and *W50*.
8. $T_3$ : Tippett's combination test of *RMD*, *W50* and *OB*.
9. $T_2$ : Tippett's combination test of *RMD* and *W50*.

### 5.2. Sample Sizes and Differences in Scale Parameters

Both equal and unequal sample size settings were examined for five treatments. For equal sample size cases, $n_i = 10$ and $n_i = 30$ were used. Unequal sample size cases, settings of $n_1 = 5$, $n_2 = 5$, $n_3 = 10$, $n_4 = 15$, $n_5 = 15$ and $n_1 = 10$, $n_2 = 10$, $n_3 = 20$, $n_4 = 30$, $n_5 = 30$ were utilized. Maximum scale parameter ratios, $\frac{\sigma_{\{max\}}}{\sigma_{\{min\}}}$, ranging from 1 to 5 were examined, with different patterns of smaller ratios present. Settings of $(\sigma, 1, 1, 1, 1)$ and $(\sigma, (\sigma+1)/2, 1, 1, 1)$ were used. The first setting we refer to as the "single extreme scale parameter" setting, while the second setting has an intermediate scale value midway between the minimum (1) and maximum ($\sigma$). The specific settings used for $(\sigma_1, \sigma_2, \sigma_3, \sigma_4, \sigma_5)$ were as follows:

1. 1.$(1, 1, 1, 1, 1)$ 2.$(3, 1, 1, 1, 1)$ 3.$(3, 2, 1, 1, 1)$ 4.$(5, 1, 1, 1, 1)$ 5.$(5, 3, 1, 1, 1)$.

### 5.3. Distributions

Several different $g$ and $h$ distributions [18] were used to simulate data from distributions with different characteristics. $g$ and $h$ distributions are monotonic functions of normal distributions and allow investigation of nonnormal distributions with specific characteristics. The $g$-and-$h$ random variable is defined as $Y_{g,h}(Z) = \left(\frac{\exp(gZ)-1}{g}\right)\exp\left(\frac{hZ^2}{2}\right)$, where $Z \sim N(0, 1)$. When $g = h = 0$, $Y_{g,h}(Z) \sim N(0, 1)$. Nonzero values of $g$ increase the skewness and positive values of $h$ increase the elongation (tail heaviness) of the distribution. Changing the values of $g$ and $h$ does not affect the location of the distribution. The following cases were considered, and representative plots shown in Figure 1:

1. $g = 0$, $h = 0$—Normally distributed (symmetric, light tails);
2. $g = 0$, $h = 0.4$—Symmetric, moderately heavy tails;
3. $g = 0$, $h = 0.8$—Symmetric, very heavy tails;
4. $g = 0.4$, $h = 0$—Moderately skewed, light tails;
5. $g = 0.8$, $h = 0$—Heavily skewed, light tails;
6. $g = 0.4$, $h = 0.4$—Moderately skewed, moderately heavy tails;
7. $g = 0.8$, $h = 0.4$—Heavily skewed, moderately heavy tails.

Type I error rate and power were estimated based on 1000 randomly selected data sets from each distribution, for each setting of sample sizes and scale parameter patterns. It has been suggested [19] that only 253 random permutations are necessary with 1000 random data sets if the goal of the simulation is to estimate the power of a test and only a "rough" estimate of the permutation $p$-value is required, while a random sample of at least 1600 permutations was recommended [20] to estimate the exact $p$-value for a permutation test. Since precise estimation of the permutation test $p$-values was considered important, a conservative sample of 1999 random permutations was utilized, and thus the permutation distribution for each test was based on 2000 values: the observed test statistic value plus 1999 values based on random permutations of the observed data.

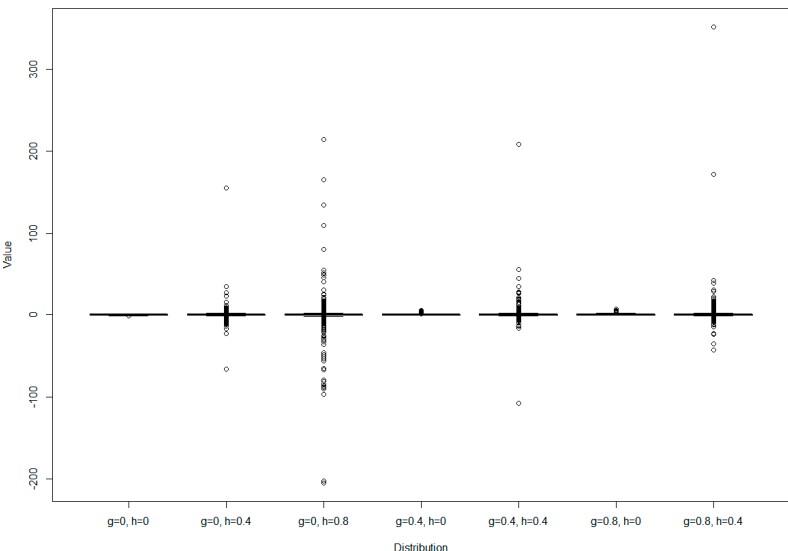

**Figure 1.** Example boxplots of the simulated distributions. Note that the "Value" axis has been truncated to omit extreme values from distributions 3 and 7.

## 6. Simulation Results

### 6.1. Familywise Type I Error

All tests were robust in the sense that estimated rates of Type I error were close to the nominal level of 0.05 (See Tables 1–6) with only one exceeding 0.075 (0.084 for *RMD* in the equal sample $n_i = 30$ case, $g = 0.8$, $h = 0.4$). Note that in the tables, the first row of each distribution represents the equal scale case, and thus the value given is the estimated Type I error rate.

### 6.2. Any-Pair Power

When sample sizes were equal (Tables 1 and 2), *RMD* tended to have the highest power, although in some cases the Fisher or Liptak combined test was most powerful.

When sample sizes were small and unequal and the *larger scales were associated with the smaller samples* (Table 3), the $F_2$ and $L_2$ combined tests were most powerful for all scale configurations, with $L_2$ usually having the higher power. The lone exception was when the distribution was symmetric with very heavy tails ($g = 0$, $h = 0.8$) where the *RMD* had similar power to $F_2$ and $L_2$. When the sample sizes increased to $n_i = 10, 10, 20, 30, 30$, however, the power advantage of the combined tests over *RMD* tended to diminish, except for the skewed, light-tailed distributions, where the combined tests were still more powerful (See Table 4).

Neither of the Tippett combined tests was as powerful as the Liptak and Fisher versions.

When the sample sizes were small and unequal but the *larger scales were associated with the larger samples* (Tables 5 and 6), $L_2$ and $F_2$ had the highest power for normal and moderately skewed-only distributions. Meanwhile, *RMD* had the highest power for all distributions with heavy tails ($h = 0.4, 0.8$). As before, as sample sizes increased, the power advantages of the combined tests diminished while *RMD* maintained power advantages for heavier-tailed distributions.

**Table 1.** Proportion of at least one rejection at $\alpha = 0.05$, five treatments, equal samples of size $n_i = 10$.

| Distribution | Scale $(\sigma_1\sigma_2\sigma_3\sigma_4\sigma_5)$ | Method | | | | | | | | |
|---|---|---|---|---|---|---|---|---|---|---|
| | | *W50* | *OB* | *RMD* | *F₃* | *F₂* | *L₃* | *L₂* | *T₃* | *T₂* |
| $g = 0, h = 0$ | 11111 | 0.040 | 0.016 | 0.039 | 0.039 | 0.042 | 0.041 | 0.043 | 0.025 | 0.034 |
| | 31111 | 0.669 | 0.620 | **0.742** | 0.708 | 0.726 | 0.708 | 0.727 | 0.689 | 0.710 |
| | 32111 | 0.609 | 0.466 | 0.689 | 0.648 | 0.692 | 0.655 | **0.697** | 0.604 | 0.632 |
| | 51111 | 0.911 | 0.850 | **0.965** | 0.935 | 0.954 | 0.933 | 0.950 | 0.944 | 0.957 |
| | 53111 | 0.889 | 0.697 | **0.970** | 0.925 | 0.956 | 0.924 | 0.957 | 0.929 | 0.944 |
| $g = 0, h = 0.4$ | 11111 | 0.010 | 0.000 | 0.065 | 0.020 | 0.034 | 0.014 | 0.031 | 0.030 | 0.043 |
| | 31111 | 0.099 | 0.028 | **0.244** | 0.155 | 0.205 | 0.126 | 0.204 | 0.191 | 0.202 |
| | 32111 | 0.084 | 0.016 | **0.274** | 0.141 | 0.212 | 0.112 | 0.205 | 0.197 | 0.225 |
| | 51111 | 0.285 | 0.090 | **0.476** | 0.378 | 0.445 | 0.331 | 0.446 | 0.411 | 0.433 |
| | 53111 | 0.192 | 0.046 | **0.495** | 0.332 | 0.434 | 0.247 | 0.421 | 0.409 | 0.433 |
| $g = 0, h = 0.8$ | 11111 | 0.004 | 0.000 | 0.071 | 0.014 | 0.031 | 0.005 | 0.024 | 0.036 | 0.051 |
| | 31111 | 0.017 | 0.003 | **0.140** | 0.057 | 0.098 | 0.032 | 0.084 | 0.092 | 0.110 |
| | 32111 | 0.011 | 0.001 | **0.182** | 0.060 | 0.110 | 0.034 | 0.088 | 0.108 | 0.133 |
| | 51111 | 0.053 | 0.011 | **0.237** | 0.130 | 0.192 | 0.079 | 0.174 | 0.189 | 0.208 |
| | 53111 | 0.034 | 0.007 | **0.286** | 0.129 | 0.208 | 0.070 | 0.182 | 0.214 | 0.240 |
| $g = 0.4, h = 0$ | 11111 | 0.042 | 0.020 | 0.049 | 0.037 | 0.053 | 0.036 | 0.056 | 0.031 | 0.037 |
| | 31111 | 0.620 | 0.552 | 0.656 | 0.660 | 0.681 | 0.665 | **0.686** | 0.627 | 0.630 |
| | 32111 | 0.549 | 0.402 | **0.637** | 0.597 | 0.630 | 0.599 | **0.638** | 0.553 | 0.585 |
| | 51111 | 0.891 | 0.816 | **0.942** | 0.933 | **0.943** | 0.928 | **0.942** | 0.931 | 0.935 |
| | 53111 | 0.842 | 0.614 | **0.931** | 0.900 | 0.928 | 0.889 | 0.928 | 0.889 | 0.906 |
| $g = 0.4, h = 0.4$ | 11111 | 0.007 | 0.001 | 0.062 | 0.015 | 0.029 | 0.011 | 0.028 | 0.023 | 0..030 |
| | 31111 | 0.090 | 0.028 | **0.240** | 0.146 | 0.204 | 0.118 | 0.200 | 0.184 | 0.200 |
| | 32111 | 0.081 | 0.014 | **0.272** | 0.130 | 0.216 | 0.099 | 0.201 | 0.195 | 0.222 |
| | 51111 | 0.251 | 0.090 | **0.442** | 0.343 | 0.418 | 0.301 | 0.415 | 0.384 | 0.406 |
| | 53111 | 0.182 | 0.040 | **0.477** | 0.310 | 0.406 | 0.248 | 0.390 | 0.374 | 0.406 |
| $g = 0.8, h = 0$ | 11111 | 0.034 | 0.014 | 0.058 | 0.042 | 0.047 | 0.036 | 0.052 | 0.033 | 0.042 |
| | 31111 | 0.426 | 0.359 | 0.472 | 0.514 | 0.490 | **0.513** | 0.497 | 0.465 | 0.458 |
| | 32111 | 0.346 | 0.236 | **0.468** | 0.432 | 0.451 | 0.430 | 0.459 | 0.413 | 0.413 |
| | 51111 | 0.765 | 0.658 | 0.800 | **0.835** | 0.826 | 0.826 | 0.828 | 0.813 | 0.802 |
| | 53111 | 0.640 | 0.426 | **0.801** | 0.780 | 0.791 | 0.754 | **0.799** | 0.755 | 0.762 |
| $g = 0.8, h = 0.4$ | 11111 | 0.011 | 0.002 | 0.063 | 0.015 | 0.030 | 0.009 | 0.027 | 0.028 | 0.038 |
| | 31111 | 0.074 | 0.025 | **0.195** | 0.138 | 0.168 | 0.104 | 0.164 | 0.154 | 0.171 |
| | 32111 | 0.064 | 0.014 | **0.231** | 0.131 | 0.179 | 0.089 | 0.167 | 0.156 | 0.185 |
| | 51111 | 0.186 | 0.065 | **0.384** | 0.294 | 0.348 | 0.242 | 0.346 | 0.313 | 0.338 |
| | 53111 | 0.138 | 0.032 | **0.430** | 0.272 | 0.356 | 0.204 | 0.346 | 0.333 | 0.364 |

**Table 2.** Proportion of at least one rejection at $\alpha = 0.05$, five treatments, equal samples of size $n_i = 30$. Cases that were uninformative for comparing methods were omitted.

| Distribution | Scale $(\sigma_1\sigma_2\sigma_3\sigma_4\sigma_5)$ | Method | | | | | | | | |
|---|---|---|---|---|---|---|---|---|---|---|
| | | *W50* | *OB* | *RMD* | *F₃* | *F₂* | *L₃* | *L₂* | *T₃* | *T₂* |
| $g = 0, h = 0$ | 11111 | 0.044 | 0.023 | 0.045 | 0.041 | 0.047 | 0.042 | 0.047 | 0.038 | 0.041 |
| $g = 0, h = 0.4$ | 11111 | 0.007 | 0.001 | 0.060 | 0.014 | 0.030 | 0.012 | 0.028 | 0.030 | 0.034 |
| | 31111 | 0.276 | 0.076 | **0.419** | 0.318 | 0.392 | 0.272 | 0.395 | 0.372 | 0.389 |
| | 32111 | 0.225 | 0.050 | **0.450** | 0.309 | 0.402 | 0.242 | 0.399 | 0.384 | 0.401 |
| | 51111 | 0.619 | 0.224 | 0.702 | 0.643 | **0.716** | 0.585 | **0.715** | 0.681 | 0.692 |
| | 53111 | 0.461 | 0.134 | 0.730 | 0.601 | 0.706 | 0.512 | 0.704 | 0.684 | 0.698 |

**Table 2.** *Cont.*

| Distribution | Scale ($\sigma_1\sigma_2\sigma_3\sigma_4\sigma_5$) | Method | | | | | | | | |
|---|---|---|---|---|---|---|---|---|---|---|
| | | *W50* | *OB* | *RMD* | *F₃* | *F₂* | *L₃* | *L₂* | *T₃* | *T₂* |
| $g = 0, h = 0.8$ | 11111 | 0.002 | 0.000 | 0.075 | 0.013 | 0.029 | 0.009 | 0.019 | 0.033 | 0.043 |
| | 31111 | 0.026 | 0.007 | **0.155** | 0.074 | 0.112 | 0.053 | 0.097 | 0.116 | 0.132 |
| | 32111 | 0.023 | 0.003 | **0.199** | 0.070 | 0.124 | 0.042 | 0.103 | 0.143 | 0.167 |
| | 51111 | 0.078 | 0.015 | **0.278** | 0.183 | 0.241 | 0.124 | 0.228 | 0.241 | 0.252 |
| | 53111 | 0.059 | 0.008 | **0.339** | 0.187 | 0.274 | 0.110 | 0.245 | 0.289 | 0.299 |
| $g = 0.4, h = 0$ | 11111 | 0.029 | 0.020 | 0.049 | 0.037 | 0.044 | 0.034 | 0.044 | 0.033 | 0.040 |
| $g = 0.4, h = 0.4$ | 11111 | 0.004 | 0.001 | 0.073 | 0.014 | 0.030 | 0.011 | 0.022 | 0.035 | 0.045 |
| | 31111 | 0.222 | 0.068 | **0.380** | 0.280 | 0.349 | 0.228 | 0.346 | 0.332 | 0.352 |
| | 32111 | 0.168 | 0.041 | **0.409** | 0.264 | 0.349 | 0.199 | 0.340 | 0.338 | 0.360 |
| | 51111 | 0.533 | 0.174 | 0.635 | 0.561 | **0.660** | 0.506 | **0.664** | 0.613 | 0.628 |
| | 53111 | 0.394 | 0.099 | **0.671** | 0.546 | 0.642 | 0.448 | 0.636 | 0.613 | 0.634 |
| $g = 0.8, h = 0$ | 11111 | 0.026 | 0.016 | 0.053 | 0.034 | 0.038 | 0.034 | 0.039 | 0.039 | 0.043 |
| | 31111 | 0.935 | 0.793 | 0.921 | 0.917 | **0.942** | 0.903 | **0.943** | 0.913 | 0.923 |
| | 32111 | 0.861 | 0.584 | 0.912 | 0.865 | 0.919 | 0.852 | **0.923** | 0.862 | 0.892 |
| $g = 0.8, h = 0.4$ | 11111 | 0.005 | 0.001 | 0.084 | 0.012 | 0.035 | 0.009 | 0.031 | 0.036 | 0.050 |
| | 31111 | 0.138 | 0.046 | **0.303** | 0.215 | 0.265 | 0.170 | 0.263 | 0.256 | 0.276 |
| | 32111 | 0.103 | 0.024 | **0.336** | 0.200 | 0.274 | 0.154 | 0.262 | 0.267 | 0.296 |
| | 51111 | 0.371 | 0.118 | **0.526** | 0.456 | 0.514 | 0.392 | 0.517 | 0.490 | 0.507 |
| | 53111 | 0.264 | 0.067 | **0.573** | 0.436 | 0.533 | 0.325 | 0.528 | 0.510 | 0.531 |

**Table 3.** Proportion of at least one rejection at $\alpha = 0.05$, five treatments, unequal samples of size $n_i = 5, 5, 10, 15, 15$, larger scale associated with smaller sample size. Cases that were uninformative for comparing methods were omitted.

| Distribution | Scales ($\sigma_1\sigma_2\sigma_3\sigma_4\sigma_5$) | Method | | | | | | | | |
|---|---|---|---|---|---|---|---|---|---|---|
| | | *W50* | *OB* | *RMD* | *F₃* | *F₂* | *L₃* | *L₂* | *T₃* | *T₂* |
| $g = 0, h = 0$ | 11111 | 0.046 | 0.006 | 0.022 | 0.025 | 0.038 | 0.027 | 0.038 | 0.017 | 0.024 |
| | 31111 | 0.331 | 0.288 | 0.128 | 0.322 | 0.294 | **0.336** | 0.301 | 0.268 | 0.258 |
| | 32111 | 0.320 | 0.222 | 0.048 | 0.302 | 0.275 | **0.317** | 0.297 | 0.235 | 0.232 |
| | 51111 | 0.502 | 0.451 | 0.348 | 0.543 | 0.525 | **0.557** | 0.546 | 0.470 | 0.460 |
| | 53111 | 0.507 | 0.330 | 0.244 | 0.512 | 0.541 | 0.546 | **0.575** | 0.413 | 0.416 |
| $g = 0, h = 0.4$ | 11111 | 0.006 | 0.004 | 0.047 | 0.007 | 0.026 | 0.005 | 0.023 | 0.022 | 0.029 |
| | 31111 | 0.053 | 0.024 | 0.060 | 0.052 | 0.075 | 0.056 | **0.077** | 0.048 | 0.053 |
| | 32111 | 0.050 | 0.023 | 0.056 | 0.051 | 0.069 | 0.051 | **0.083** | 0.030 | 0.039 |
| | 51111 | 0.120 | 0.053 | 0.132 | 0.144 | 0.186 | 0.140 | **0.195** | 0.109 | 0.124 |
| | 53111 | 0.120 | 0.046 | 0.114 | 0.137 | 0.184 | 0.141 | **0.196** | 0.093 | 0.122 |
| $g = 0, h = 0.8$ | 11111 | 0.003 | 0.003 | 0.068 | 0.006 | 0.023 | 0.006 | 0.017 | 0.026 | 0.037 |
| $g = 0.8, h = 0$ | 11111 | 0.027 | 0.005 | 0.032 | 0.019 | 0.039 | 0.018 | 0.042 | 0.020 | 0.032 |
| | 31111 | 0.217 | 0.181 | 0.090 | **0.220** | 0.210 | 0.222 | **0.217** | 0.181 | 0.177 |
| | 32111 | 0.224 | 0.141 | 0.040 | **0.221** | 0.205 | 0.227 | **0.223** | 0.174 | 0.165 |
| | 51111 | 0.409 | 0.318 | 0.216 | **0.431** | 0.415 | 0.430 | **0.433** | 0.365 | 0.370 |
| | 53111 | 0.421 | 0.251 | 0.143 | 0.402 | 0.428 | 0.415 | **0.453** | 0.327 | 0.344 |
| $g = 0.4, h = 0$ | 11111 | 0.036 | 0.009 | 0.024 | 0.026 | 0.042 | 0.027 | 0.044 | 0.018 | 0.028 |
| | 31111 | 0.306 | 0.257 | 0.114 | 0.286 | 0.280 | **0.304** | 0.282 | 0.232 | 0.234 |
| | 32111 | 0.304 | 0.196 | 0.044 | 0.274 | 0.270 | **0.291** | 0.287 | 0.220 | 0.221 |
| | 51111 | 0.483 | 0.415 | 0.306 | 0.523 | 0.501 | **0.532** | 0.519 | 0.434 | 0.436 |
| | 53111 | 0.421 | 0.277 | 0.180 | 0.423 | 0.427 | 0.440 | **0.461** | 0.345 | 0.353 |

**Table 3.** *Cont.*

| Distribution | Scales $(\sigma_1\sigma_2\sigma_3\sigma_4\sigma_5)$ | Method | | | | | | | | |
|---|---|---|---|---|---|---|---|---|---|---|
| | | *W50* | *OB* | *RMD* | *F₃* | *F₂* | *L₃* | *L₂* | *T₃* | *T₂* |
| $g = 0.4, h = 0.4$ | 11111 | 0.007 | 0.001 | 0.056 | 0.005 | 0.023 | 0.007 | 0.023 | 0.027 | 0.032 |
| | 31111 | 0.046 | 0.022 | 0.061 | 0.058 | 0.065 | 0.051 | **0.074** | 0.047 | 0.051 |
| | 32111 | 0.054 | 0.019 | 0.054 | 0.050 | 0.073 | 0.053 | **0.082** | 0.046 | 0.050 |
| | 51111 | 0.116 | 0.054 | 0.127 | 0.132 | 0.169 | 0.123 | **0.180** | 0.108 | 0.124 |
| | 53111 | 0.084 | 0.040 | 0.070 | 0.089 | 0.114 | 0.094 | **0.127** | 0.069 | 0.077 |
| $g = 0.8, h = 0.4$ | 11111 | 0.006 | 0.003 | 0.064 | 0.004 | 0.027 | 0.004 | 0.020 | 0.033 | 0.035 |
| | 31111 | 0.037 | 0.017 | 0.052 | 0.047 | 0.058 | 0.045 | **0.063** | 0.039 | 0.041 |
| | 32111 | 0.045 | 0.020 | 0.053 | 0.051 | 0.073 | 0.049 | **0.074** | 0.045 | 0.057 |
| | 51111 | 0.104 | 0.044 | 0.126 | 0.116 | 0.147 | 0.108 | **0.159** | 0.097 | 0.119 |
| | 53111 | 0.097 | 0.040 | 0.104 | 0.113 | 0.146 | 0.118 | **0.156** | 0.097 | 0.107 |

**Table 4.** Proportion of at least one rejection at $\alpha = 0.05$, five treatments, unequal samples of size $n_i = 10, 10, 20, 30, 30$, larger scale associated with smaller sample size.

| Distribution | Scales $(\sigma_1\sigma_2\sigma_3\sigma_4\sigma_5)$ | Method | | | | | | | | |
|---|---|---|---|---|---|---|---|---|---|---|
| | | *W50* | *OB* | *RMD* | *F₃* | *F₂* | *L₃* | *L₂* | *T₃* | *T₂* |
| $g = 0, h = 0$ | 11111 | 0.039 | 0.014 | 0.029 | 0.034 | 0.036 | 0.034 | 0.038 | 0.024 | 0.030 |
| | 31111 | 0.813 | 0.777 | 0.771 | **0.830** | 0.816 | **0.834** | 0.823 | 0.770 | 0.769 |
| | 32111 | 0.735 | 0.591 | 0.746 | 0.792 | 0.789 | **0.800** | **0.806** | 0.667 | 0.687 |
| | 51111 | 0.965 | 0.920 | 0.977 | 0.974 | 0.984 | 0.975 | **0.984** | 0.963 | 0.967 |
| | 53111 | 0.948 | 0.769 | **0.990** | 0.975 | 0.987 | 0.975 | **0.989** | 0.960 | 0.972 |
| $g = 0, h = 0.4$ | 11111 | 0.005 | 0.002 | 0.049 | 0.010 | 0.019 | 0.008 | 0.017 | 0.022 | 0.030 |
| | 31111 | 0.105 | 0.033 | **0.176** | 0.134 | **0.179** | 0.124 | **0.181** | 0.129 | 0.145 |
| | 32111 | 0.063 | 0.026 | **0.181** | 0.106 | 0.158 | 0.092 | **0.176** | 0.113 | 0.142 |
| | 51111 | 0.296 | 0.110 | **0.412** | 0.336 | **0.417** | 0.322 | **0.422** | 0.350 | 0.374 |
| | 53111 | 0.184 | 0.062 | **0.410** | 0.289 | 0.399 | 0.263 | **0.412** | 0.316 | 0.346 |
| $g = 0, h = 0.8$ | 11111 | 0.003 | 0.001 | 0.062 | 0.009 | 0.022 | 0.005 | 0.013 | 0.028 | 0.034 |
| | 31111 | 0.016 | 0.008 | **0.064** | 0.029 | 0.055 | 0.023 | 0.054 | 0.036 | 0.044 |
| | 32111 | 0.011 | 0.007 | **0.074** | 0.021 | 0.050 | 0.019 | 0.046 | 0.040 | 0.049 |
| | 51111 | 0.041 | 0.012 | **0.122** | 0.080 | 0.114 | 0.069 | 0.117 | 0.089 | 0.106 |
| | 53111 | 0.026 | 0.008 | **0.122** | 0.063 | 0.100 | 0.060 | 0.104 | 0.086 | 0.096 |
| $g = 0.8, h = 0$ | 11111 | 0.030 | 0.009 | 0.035 | 0.030 | 0.039 | 0.031 | 0.041 | 0.025 | 0.030 |
| | 31111 | 0.489 | 0.391 | 0.443 | 0.525 | 0.525 | **0.529** | **0.536** | 0.479 | 0.449 |
| | 32111 | 0.378 | 0.258 | 0.389 | 0.445 | 0.451 | 0.446 | **0.472** | 0.371 | 0.366 |
| | 51111 | 0.843 | 0.710 | 0.827 | 0.863 | 0.860 | 0.862 | **0.865** | 0.832 | 0.826 |
| | 53111 | 0.680 | 0.477 | 0.803 | 0.795 | 0.833 | 0.786 | **0.840** | 0.726 | 0.758 |
| $g = 0.4, h = 0$ | 11111 | 0.007 | 0.001 | 0.055 | 0.012 | 0.027 | 0.013 | 0.020 | 0.026 | 0.032 |
| | 31111 | 0.732 | 0.687 | 0.682 | 0.763 | 0.742 | **0.769** | 0.753 | 0.714 | 0.686 |
| | 32111 | 0.633 | 0.484 | 0.630 | 0.679 | 0.699 | 0.690 | **0.719** | 0.584 | 0.590 |
| | 51111 | 0.948 | 0.887 | 0.955 | **0.961** | 0.959 | **0.963** | 0.965 | 0.942 | 0.946 |
| | 53111 | 0.886 | 0.686 | 0.956 | 0.936 | **0.960** | 0.936 | **0.965** | 0.918 | 0.928 |
| $g = 0.4, h = 0.4$ | 11111 | 0.019 | 0.007 | 0.045 | 0.021 | 0.035 | 0.020 | 0.038 | 0.023 | 0.026 |
| | 31111 | 0.080 | 0.039 | **0.161** | 0.117 | **0.159** | 0.111 | **0.161** | 0.120 | 0.137 |
| | 32111 | 0.056 | 0.023 | **0.166** | 0.099 | 0.146 | 0.089 | 0.155 | 0.101 | 0.130 |
| | 51111 | 0.265 | 0.097 | 0.375 | 0.316 | 0.383 | 0.293 | **0.384** | 0.307 | 0.339 |
| | 53111 | 0.169 | 0.059 | **0.382** | 0.277 | 0.371 | 0.259 | 0.375 | 0.285 | 0.315 |
| $g = 0.8, h = 0.4$ | 11111 | 0.006 | 0.001 | 0.064 | 0.011 | 0.027 | 0.010 | 0.024 | 0.029 | 0.031 |
| | 31111 | 0.060 | 0.025 | **0.128** | 0.095 | 0.116 | 0.087 | 0.120 | 0.091 | 0.103 |
| | 32111 | 0.043 | 0.022 | **0.140** | 0.075 | 0.118 | 0.075 | 0.122 | 0.084 | 0.097 |
| | 51111 | 0.189 | 0.063 | 0.287 | 0.242 | 0.296 | 0.232 | **0.300** | 0.238 | 0.255 |
| | 53111 | 0.113 | 0.045 | **0.299** | 0.213 | 0.274 | 0.195 | 0.280 | 0.221 | 0.248 |

**Table 5.** Proportion of at least one rejection at $\alpha = 0.05$, five treatments, unequal samples of size $n_i = 15, 15, 10, 5, 5$, larger scale associated with larger sample size.

| Distribution | Scales | Method | | | | | | | | |
|---|---|---|---|---|---|---|---|---|---|---|
| | $(\sigma_1 \sigma_2 \sigma_3 \sigma_4 \sigma_5)$ | *W50* | *OB* | *RMD* | $F_3$ | $F_2$ | $L_3$ | $L_2$ | $T_3$ | $T_2$ |
| $g = 0, h = 0$ | 11111 | 0.046 | 0.006 | 0.022 | 0.025 | 0.038 | 0.027 | 0.038 | 0.017 | 0.024 |
| | 31111 | 0.539 | 0.008 | 0.493 | 0.518 | 0.614 | 0.544 | **0.655** | 0.413 | 0.469 |
| | 32111 | 0.511 | 0.003 | 0.461 | 0.468 | 0.589 | 0.504 | **0.614** | 0.383 | 0.445 |
| | 51111 | 0.781 | 0.016 | 0.915 | 0.854 | 0.929 | 0.861 | **0.940** | 0.815 | 0.850 |
| | 53111 | 0.764 | 0.011 | 0.888 | 0.818 | 0.901 | 0.822 | **0.916** | 0.780 | 0.819 |
| $g = 0, h = 0.4$ | 11111 | 0.006 | 0.004 | 0.047 | 0.007 | 0.026 | 0.005 | 0.023 | 0.022 | 0.029 |
| | 31111 | 0.014 | 0.000 | **0.215** | 0.049 | 0.131 | 0.029 | 0.116 | 0.130 | 0.147 |
| | 32111 | 0.008 | 0.000 | **0.255** | 0.032 | 0.128 | 0.022 | 0.097 | 0.145 | 0.167 |
| | 51111 | 0.038 | 0.000 | **0.455** | 0.153 | 0.333 | 0.091 | 0.284 | 0.327 | 0.357 |
| | 53111 | 0.019 | 0.000 | **0.490** | 0.127 | 0.316 | 0.054 | 0.240 | 0.343 | 0.374 |
| $g = 0, h = 0.8$ | 11111 | 0.003 | 0.003 | 0.068 | 0.006 | 0.023 | 0.006 | 0.017 | 0.026 | 0.037 |
| | 31111 | 0.000 | 0.000 | **0.127** | 0.110 | 0.058 | 0.003 | 0.025 | 0.079 | 0.092 |
| | 32111 | 0.000 | 0.002 | **0.172** | 0.012 | 0.012 | 0.070 | 0.002 | 0.027 | 0.105 |
| | 51111 | 0.002 | 0.000 | **0.238** | 0.052 | 0.148 | 0.009 | 0.078 | 0.170 | 0.190 |
| | 53111 | 0.000 | 0.001 | **0.314** | 0.045 | 0.163 | 0.004 | 0.058 | 0.212 | 0.241 |
| $g = 0.8, h = 0$ | 11111 | 0.027 | 0.005 | 0.032 | 0.019 | 0.039 | 0.018 | 0.042 | 0.020 | 0.032 |
| | 31111 | 0.213 | 0.004 | 0.364 | 0.250 | 0.382 | 0.231 | **0.403** | 0.269 | 0.313 |
| | 32111 | 0.161 | 0.000 | **0.375** | 0.217 | 0.361 | 0.188 | 0.366 | 0.269 | 0.306 |
| | 51111 | 0.370 | 0.013 | 0.690 | 0.563 | 0.728 | 0.505 | **0.739** | 0.616 | 0.664 |
| | 53111 | 0.281 | 0.004 | **0.691** | 0.486 | 0.682 | 0.392 | 0.684 | 0.582 | 0.627 |
| $g = 0.4, h = 0$ | 11111 | 0.036 | 0.009 | 0.024 | 0.026 | 0.042 | 0.027 | 0.044 | 0.018 | 0.028 |
| | 31111 | 0.442 | 0.006 | 0.450 | 0.440 | 0.566 | 0.442 | **0.595** | 0.395 | 0.449 |
| | 32111 | 0.380 | 0.004 | 0.436 | 0.378 | 0.521 | 0.373 | **0.544** | 0.352 | 0.409 |
| | 51111 | 0.652 | 0.015 | 0.853 | 0.774 | 0.899 | 0.757 | **0.909** | 0.755 | 0.796 |
| | 53111 | 0.605 | 0.006 | 0.832 | 0.714 | 0.856 | 0.680 | **0.867** | 0.714 | 0.759 |
| $g = 0.4, h = 0.4$ | 11111 | 0.007 | 0.001 | 0.056 | 0.005 | 0.023 | 0.007 | 0.023 | 0.027 | 0.032 |
| | 31111 | 0.007 | 0.000 | **0.214** | 0.050 | 0.134 | 0.031 | 0.101 | 0.138 | 0.156 |
| | 32111 | 0.007 | 0.001 | **0.249** | 0.035 | 0.136 | 0.019 | 0.098 | 0.148 | 0.180 |
| | 51111 | 0.030 | 0.000 | **0.430** | 0.158 | 0.314 | 0.090 | 0.275 | 0.315 | 0.337 |
| | 53111 | 0.014 | 0.000 | 0.479 | 0.133 | 0.303 | 0.056 | 0.231 | 0.331 | 0.366 |
| $g = 0.8, h = 0.4$ | 11111 | 0.006 | 0.003 | 0.064 | 0.004 | 0.027 | 0.004 | 0.020 | 0.033 | 0.035 |
| | 31111 | 0.006 | 0.000 | **0.211** | 0.036 | 0.125 | 0.018 | 0.089 | 0.133 | 0.149 |
| | 32111 | 0.003 | 0.001 | **0.246** | 0.031 | 0.131 | 0.016 | 0.074 | 0.156 | 0.179 |
| | 51111 | 0.017 | 0.000 | **0.377** | 0.137 | 0.283 | 0.057 | 0.236 | 0.301 | 0.302 |
| | 53111 | 0.010 | 0.001 | **0.437** | 0.119 | 0.288 | 0.035 | 0.197 | 0.330 | 0.362 |

**Table 6.** Proportion of at least one rejection at $\alpha = 0.05$, five treatments, unequal samples of size $n_i = 30, 30, 20, 10, 10$, larger scale associated with larger sample size. Cases that were uninformative for comparing methods were omitted.

| Distribution | Scales | Method | | | | | | | | |
|---|---|---|---|---|---|---|---|---|---|---|
| | $(\sigma_1 \sigma_2 \sigma_3 \sigma_4 \sigma_5)$ | *W50* | *OB* | *RMD* | $F_3$ | $F_2$ | $L_3$ | $L_2$ | $T_3$ | $T_2$ |
| $g = 0, h = 0$ | 11111 | 0.039 | 0.014 | 0.029 | 0.034 | 0.036 | 0.034 | 0.038 | 0.024 | 0.030 |
| | 31111 | 0.959 | 0.361 | **0.977** | 0.968 | **0.983** | 0.971 | **0.983** | 0.946 | 0.961 |
| | 32111 | 0.936 | 0.319 | 0.960 | 0.944 | **0.970** | 0.949 | **0.971** | 0.912 | 0.934 |
| $g = 0, h = 0.4$ | 11111 | 0.005 | 0.002 | 0.049 | 0.010 | 0.019 | 0.008 | 0.017 | 0.022 | 0.030 |
| | 31111 | 0.019 | 0.000 | **0.332** | 0.123 | 0.239 | 0.066 | 0.204 | 0.243 | 0.268 |
| | 32111 | 0.011 | 0.000 | **0.355** | 0.107 | 0.231 | 0.048 | 0.175 | 0.256 | 0.285 |
| | 51111 | 0.057 | 0.000 | **0.620** | 0.381 | 0.569 | 0.205 | 0.515 | 0.560 | 0.570 |
| | 53111 | 0.023 | 0.000 | **0.65** | 0.305 | 0.539 | 0.122 | 0.424 | 0.577 | 0.602 |

**Table 6.** *Cont.*

| Distribution | Scales $(\sigma_1\sigma_2\sigma_3\sigma_4\sigma_5)$ | Method | | | | | | | | |
|---|---|---|---|---|---|---|---|---|---|---|
| | | *W50* | *OB* | *RMD* | $F_3$ | $F_2$ | $L_3$ | $L_2$ | $T_3$ | $T_2$ |
| $g=0, h=0.8$ | 11111 | 0.003 | 0.001 | 0.062 | 0.009 | 0.022 | 0.005 | 0.013 | 0.028 | 0.034 |
| | 31111 | 0.000 | 0.000 | **0.156** | 0.024 | 0.089 | 0.004 | 0.042 | 0.113 | 0.122 |
| | 32111 | 0.000 | 0.000 | **0.210** | 0.029 | 0.104 | 0.006 | 0.038 | 0.143 | 0.163 |
| | 51111 | 0.000 | 0.000 | **0.290** | 0.095 | 0.199 | 0.013 | 0.119 | 0.232 | 0.257 |
| | 53111 | 0.000 | 0.000 | **0.369** | 0.092 | 0.230 | 0.009 | 0.116 | 0.287 | 0.314 |
| $g=0.8, h=0$ | 11111 | 0.030 | 0.009 | 0.035 | 0.030 | 0.039 | 0.031 | 0.041 | 0.025 | 0.030 |
| | 31111 | 0.372 | 0.034 | 0.662 | 0.550 | 0.677 | 0.496 | **0.695** | 0.566 | 0.617 |
| | 32111 | 0.271 | 0.007 | **0.651** | 0.458 | 0.610 | 0.382 | 0.622 | 0.512 | 0.558 |
| | 51111 | 0.607 | 0.065 | 0.952 | 0.917 | 0.970 | 0.820 | **0.970** | 0.924 | 0.936 |
| | 53111 | 0.484 | 0.020 | **0.944** | 0.857 | **0.941** | 0.716 | **0.946** | 0.905 | 0.921 |
| $g=0.4, h=0$ | 11111 | 0.007 | 0.001 | 0.055 | 0.012 | 0.027 | 0.013 | 0.020 | 0.026 | 0.032 |
| | 31111 | 0.807 | 0.170 | 0.921 | 0.889 | 0.942 | 0.876 | **0.951** | 0.864 | 0.889 |
| | 32111 | 0.748 | 0.111 | 0.892 | 0.825 | 0.903 | 0.805 | **0.911** | 0.811 | 0.844 |
| | 51111 | 0.947 | 0.242 | 0.999 | 0.997 | 0.999 | 0.989 | 0.999 | 0.996 | 0.998 |
| | 53111 | 0.936 | 0.189 | 0.998 | 0.991 | 0.999 | 0.975 | 0.999 | 0.993 | 0.993 |
| $g=0.4, h=0.4$ | 11111 | 0.019 | 0.007 | 0.045 | 0.021 | 0.035 | 0.020 | 0.038 | 0.023 | 0.026 |
| | 31111 | 0.009 | 0.000 | **0.316** | 0.098 | 0.216 | 0.040 | 0.180 | 0.228 | 0.252 |
| | 32111 | 0.005 | 0.000 | **0.349** | 0.086 | 0.217 | 0.030 | 0.171 | 0.240 | 0.266 |
| | 51111 | 0.043 | 0.000 | **0.582** | 0.346 | 0.528 | 0.180 | 0.461 | 0.507 | 0.529 |
| | 53111 | 0.015 | 0.000 | **0.627** | 0.299 | 0.511 | 0.106 | 0.396 | 0.550 | 0.575 |
| $g=0.8, h=0.4$ | 11111 | 0.006 | 0.001 | 0.064 | 0.011 | 0.027 | 0.010 | 0.024 | 0.029 | 0.031 |
| | 31111 | 0.002 | 0.000 | **0.259** | 0.071 | 0.164 | 0.023 | 0.128 | 0.188 | 0.208 |
| | 32111 | 0.001 | 0.000 | **0.301** | 0.067 | 0.167 | 0.016 | 0.112 | 0.208 | 0.235 |
| | 51111 | 0.016 | 0.000 | **0.503** | 0.277 | 0.438 | 0.115 | 0.360 | 0.434 | 0.455 |
| | 53111 | 0.005 | 0.000 | **0.569** | 0.235 | 0.432 | 0.064 | 0.310 | 0.488 | 0.518 |

## 7. Discussion

In this paper, we studied the performance of nonparametric combined tests for multiple comparisons of scale parameters. The *RMD* test had been shown in previous studies to be the preferred test to compare scale parameters, although it was not always the most powerful test, as the *W50* and *OB* tests were able to outperform *RMD* in some situations. We found that combinations of two or more of these tests could be more powerful than any individual test. Distribution and sample size configurations for which *W50* and/or *OB* were more powerful than *RMD* tended to be the cases where a combined test was found to be most powerful. Combined tests tended to outperform *RMD* for skewed, lighter-tailed distributions, while *RMD* tended to be more powerful when distributions were heavier-tailed, since in these scenarios, *RMD* enjoyed large power advantages over *W50* and *OB*. Combined tests involving *OB* never showed an advantage over combinations of only *RMD* and *W50*.

As with any simulation study, generalization of results requires caution. These results may not extend to situations where the true scales are very different than those studied here, and/or where the data do not come from the distributions studied here. In addition, the conclusions rely on the assumption of a location-scale model, and thus may not be valid if that assumption is not plausible.

**Author Contributions:** Conceptualization, S.J.R. and M.H.M.; methodology, S.J.R. and M.H.M.; software, S.J.R.; validation, S.J.R.; formal analysis, S.J.R. and M.H.M.; investigation, S.J.R.; resources, S.J.R.; data curation, S.J.R.; writing—original draft preparation, S.J.R.; writing—review and editing, S.J.R. and M.H.M.; visualization, S.J.R.; project administration, S.J.R. All authors have read and agreed to the published version of the manuscript.

**Funding:** This research received no external funding.

**Institutional Review Board Statement:** Not applicable.

**Informed Consent Statement:** Not applicable.

**Data Availability Statement:** The original contributions presented in the study are included in the article, further inquiries can be directed to the corresponding author/s.

**Conflicts of Interest:** The authors declare no conflicts of interest.

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
