# Peer review of "Combined Permutation Tests for Pairwise Comparison of Scale Parameters Using Deviances"

_stats, doi:10.3390/stats7020021_

Round 1

Reviewer 1 Report

Comments and Suggestions for Authors

This paper proposes the method of combined test using permutation. My major concern is the procedure for the permutation test for the combined p-value including RMD is not clear, since RMD already uses permutation to get the critical value (comment 2). Also, the result visualization needs improvement (comment 5).

Comments:

1. Line 111, list the reference to the past simulations.

2. I am not very clear about the combined test. Take the Fisher combining function for example, is T_F the p-value of the combined test? More details about the combined test should be added here. A flow chart for the procedure in lines 111-117 is preferred. In addition, RMD already uses permutation to get the critical values. Later when you use permutations to get the combined p-value involving RMD, is this a nested permutation?

3. Lines 155-161, it would be good to plot the pdf of these distributions to give readers a brief impression.

4. One limitation of this paper is the testing procedure is for scale parameters.

5. The tables in this paper take up too much space and are not informative. The main text should only display some typical settings and the rest of them should be moved to the supplemental material. For each row, the best method’s number should be bold.

Reviewer 2 Report

Comments and Suggestions for Authors

Please see my attached pdf report.

Round 2

Reviewer 1 Report

Comments and Suggestions for Authors

Thanks for the responses. I do not have any further comments.